# Numerical Study of Casing Microleakage Flow Field Sensitivity and Acoustic Field Characteristics

**DOI:** 10.3390/ma16010386

**Published:** 2022-12-31

**Authors:** Jingcui Li, Jifang Wan, Hangming Liu, Xianzhong Yi, Yuxian He, Kang Chen, Xinbo Zhao

**Affiliations:** 1CNPC Engineering Technology R&D Company Limited, Beijing 102206, China; 2School of Mechanical Engineering, Yangtze University, Jingzhou 434023, China; 3School of Petroleum and Natural Gas Engineering, Changzhou University, Changzhou 213164, China

**Keywords:** casing pipes, micro-leakage, ultrasonic detection, acoustic vibration coupling, numerical simulation

## Abstract

The casing leakage phenomenon seriously affects the safety and economic problems of oil and gas production and transportation. In this paper, the numerical simulation study of the casing’s micro-leakage flow field and acoustic field is carried out by taking the oil and gas well casing as the research object. The CFD numerical model of the casing micro-leakage is established, and the influence of the size of the leakage hole, the shape of the leakage hole, and the pressure difference between the inside and outside the casing on the microleakage flow field is analyzed. An acoustic-vibroacoustic coupling calculation model based on Fluent and LMS Virtual LAB is established, and the sound pressure value and distribution at different frequencies are calculated. The results show that the flow rate of the leakage hole increases with the pressure difference between the inside and the outside leakage hole and the area of the leakage hole. Moreover, the flow rate of the circular leakage hole is higher for the same leakage hole area. The simulation model based on the equivalent sound source can be used to calculate and analyze the sound field in the tubing. By sound field computation based on the near-field equivalent sound source, when the frequency is 32,000 Hz, the amplitude of sound pressure is maximum. In addition, the sound pressure is greatly reduced once the sound wave passes through the tubing pipeline. Lastly, the sound pressure is higher at the position facing the leakage hole in the tubing, making detecting the leakage sound field signal easier. The results can provide a reference for further research on oil casing microleakage detection technology.

## 1. Introduction

Natural gas is a high-quality, efficient, clean energy that has received universal attention and has been a priority in all countries worldwide. In an environment of high temperature and high pressure, natural gas pipelines are easily eroded by corrosive media, and pipeline leakage accidents may occur in long-term operations, which greatly affect people’s health and property safety [1,2,3,4,5]. Therefore, establishing pipeline leakage detection techniques and the timely detection of pipeline leakage locations can not only avoid the occurrence of malignant leakage accidents, but also ensure the safe and efficient development and production of oil and gas fields. At present, the commonly used detection methods for leakage of natural gas pipelines include the negative pressure wave method [6], distributed optical fiber method [7], and acoustic wave method [8]. Among them, the acoustic wave method can be divided into infrasonic, acoustic, and ultrasonic waves according to the different ranges of detected frequencies.

Gas well oil casing is a special type of pipeline with a more complex structure than natural gas transmission pipelines. Due to the cementing operation, cement consolidates the outer wall of the gas well’s casing. Hence, the leakage location method of the surface transportation pipeline cannot detect the leakage of the casing directly. As a result, the leakage detection of the casing can only be carried out internally. Researchers often adopt the method of acoustic leakage detection technology to improve the accuracy of casing leakage detection. In addition, high ultrasonic energy, strong orientation, and strong penetration are more suitable for leakage detection of gas well oil casing [9,10,11,12,13].

A few studies have been carried out on ultrasonic detection of oil pipe leakage, but relatively few focus on casing pipe leakage. Huang et al. [14] studied the gas diffusion distribution characteristics of rectangular leakage holes and square leakage holes by the leakage model of natural gas pipelines. Based on the FW-H model, Han et al. [15] computed the flow field distribution characteristics of leakage holes with equal cross-sectional area but different shapes under the same working condition. Moreover, the authors analyzed the influences of the shapes of leakage holes on the flow field characteristics of the micro-leakage process, which provided the theoretical basis and data support for the study of the flow field and sound source characteristics of pipeline micro-leakage. Through computational fluid dynamics (CFD) fluid simulation, Zhang et al. [16] analyzed the influences of the shape of leakage holes, the size of leakage holes, the internal pressure of oil casing pipes, and the annulus medium on the leakage flow field. According to their results, the circular leakage hole had the largest leakage flow rate, and the gap leakage hole had the smallest; the larger the leakage hole, the greater the pressure difference inside and outside the oil casing pipes. Mori et al. [17] carried out simulation experiments on the aeroacoustics and acoustics of T-shaped rectangular cross-section pipelines under different inlet flow rates, and the experimental results showed that the sound source characteristics of aerodynamic noise generated by the flow in the pipelines were affected by the flow rate of the fluid inside the pipelines. Sun et al. [18] analyzed the sound source characteristics of the leakage of casing pipes, established a simulation model for the leakage of casing pipes in gas wells, and numerically simulated the flow field and sound field of the leakage of casing pipes. Martins et al. [19] established a three-dimensional pipeline leakage model. They analyzed the transient flow and pressure changes of high-pressure pipeline leakage through fluid simulation and test verification, especially the flow state of the central axis and hole wall of leakage holes. Using fluid theory, Zhang et al. [20] established a mathematical model of small hole leakage. The authors revealed the leakage rules of small and micro holes on natural gas pipelines under different environments, which helped improve the safety of pipeline transportation. Liu et al. [21] studied the flow field and sound field distribution and the sound mechanism of pipeline leakage by combining numerical simulation and field experiments. Moreover, the authors pointed out that the acoustic wave method was an efficient and accurate method for pipeline leakage detection. Doshmanziari et al. [22] proposed a model-based leakage detection and location method for finite-length pipelines, conducted numerical simulation for high-pressure pipeline leakage using OLGA (version of 2017.2.0) multiphase flow simulation software, and experimentally verified the rationality of the proposed method.

In summary, the existing research on pipeline micro-leakage mainly focuses on the diffusion distribution of gas after a high-pressure gas pipeline leaks, the influence of leakage pore size, pipeline pressure convection field, and acoustic detection methods and principles. In the research on propagation characteristics of sound waves and the acquisition and processing of signals, insufficient attention is paid to the mechanism of pipeline leakage sound waves. This is particularly true for the influence of different convection fields and sound source characteristics of micro-leakage hole cross-sectional area and shape. Simulating pipeline leakage is mostly carried out by simulating the flow field, mainly studying the influence of the leakage pore size and pipeline pressure on the flow field and the diffusion distribution of gas after leakage. However, sound field simulation of pipeline leakage is less often carried out. Hence, a numerical model of casing’s micro-leakage CFD is established to solve the above problems. The effect of leakage hole size, shape, and pressure difference inside and outside the leakage hole on the microleakage flow field is analyzed. An acoustic-vibroacoustic coupling calculation model based on Fluent and LMS Virtual LAB was established, and the sound pressure value and distribution at different frequencies were calculated. The results provide a reference for further research on oil casing microleakage detection technology.

## 2. Problem Description and Theoretical Analysis

### 2.1. Problem Description

Figure 1 illustrates the schematic of the physical model for the micro-leakage of casing pipes. The medium in the pipes is methane, the annular medium is air, and the environment outside the annular medium is assumed to be air with a pressure of 0.1 MPa. When casing pipes leaks, high-pressure gas in the annular air is ejected from the micro-holes in the casing wall and produces a sound source. The sound source signals can be detected by the ultrasonic instrument arranged in the casing pipes, thus achieving the purpose of pipeline leakage detection. The model parameters are listed in Table 1.

### 2.2. Theoretical Analysis

After the leakage of casing pipes occurs, fluid injection at the leakage holes can generate sound sources. The sound field first propagates through the medium in the annulus, causes the vibration of the oil pipe wall, and finally generates a sound field in the oil pipes.

According to the acoustic comparison theory in aeroacoustics, aerodynamic sound sources can be divided with respect to physical mechanisms into quadrupole sound sources, dipole sound sources, and monopole sound sources. All aerodynamic sound sources can be regarded as the combined distribution of quadrupole, dipole, and monopole [23]. In casing pipe leakage, the fluid interacts with the wall surface of the leakage holes to produce a dipole sound source, and the high-speed flow of the fluid near the leakage holes produces turbulent stress, that is, a quadrupole sound source. Monopole sound sources generally only exist in an unstable state when the gas flow rate is low. Since the flow rate near the leakage holes is relatively high, this study does not consider monopole sound sources. The ratio of the radiated sound power of the quadrupole source to that of the dipole source is proportional to the quadratic square of the gas flow rate. Here, considering that the large inner-outer pressure difference produces a high gas flow rate at the leakage holes [24,25,26], the leakage sound source can be characterized by the quadrupole sound source in the computation.

Quadrupole sound sources are originated from velocity pulsations produced by fluid leakage. The flow process of fluid can be obtained by solving the equation set of flow mechanics with the assistance of the commercial CFD solver Fluent. The propagation of the sound field in the annulus region can be obtained by solving the FW-H equation based on the acoustic comparison theory, and Fluent also contains the module for solving the FW-H equation [26]:(1)$$\frac{\partial^{2}p^{\prime }}{\partial^{2}t}-c_{0}^{2}\nabla^{2}p^{\prime }=\frac{\partial }{\partial t}\left[\rho_{0}u_{i}\frac{\partial f}{\partial x_{i}}\delta \left(f\right)\right]-\frac{\partial }{\partial x_{i}}\left[p^{\prime }\delta_{ij}\frac{\partial f}{\partial x_{i}}\delta \left(f\right)\right]+\frac{\partial^{2}T_{ij}}{\partial x_{i}\partial x_{j}}$$
where the right source terms are monopole, dipole, and quadrupole sources successively; $p^{\prime }$ is the far-field sound pressure (Pa); f is the mass force acting on the fluid per unit mass (N); $c_{0}$ is the local sound velocity in a stationary fluid (m/s); $\rho_{0}$ is the density of the stationary fluid (kg/m^3^) which satisfies $p^{\prime }=p-p_{0}$; δ(f) is the Dirac function, which satisfies the following:(2)$$H\left(f\right)=\{\begin{array}{c} 1,f\left(x,t\right)>0\\ 0,f\left(x,t\right)<0 \end{array}\;H\left(f\right)=\{\begin{array}{c} 1,f\left(x_{j},t\right)>0\\ 0,f\left(x_{i},t\right)<0 \end{array}$$
(3)$$\delta \left(f\right)=\frac{\partial H\left(f\right)}{\partial f}$$

In the leakage detection process of casing pipes in gas production wells, the ultrasonic testing equipment is usually placed inside oil pipes and separated from the leakage sound source by the oil pipes. Therefore, the coupling computation of the leakage sound field and the acoustic vibration of the oil pipeline should be considered. The acoustic vibration coupling equation is as follows [18]:(4)$$\left[\begin{array}{cc} K_{s}+i\omega D_{s}-\omega^{2}M_{s} & C\\ \omega^{2}C^{T} & K_{a}+i\omega D_{a}-\omega^{2}M_{a} \end{array}\right]\left\{\begin{array}{c} u\left(\omega \right)\\ p\left(\omega \right) \end{array}\right\}=\left\{\begin{array}{c} f_{s}\omega \\ f_{a}\omega \end{array}\right\}$$
where $K_{s}$, $D_{s}$, and $M_{s}$ are the stiffness matrix, damping matrix, and mass matrix of the solid structure, respectively; $K_{a}$, $D_{a}$, and $M_{a}$ are the stiffness matrix, damping matrix, and mass matrix of the fluid, respectively; C is the coupling matrix; $f_{s}$ is the structure load vector; $f_{a}$ is the acoustic load vector; u and p are the displacement vector and sound pressure vector, respectively.

The grid size directly affects the frequency of solving the coupling equation in acoustic finite element analysis. Generally, the side length of the maximum computing cell is less than 1/6 of the shortest wavelength [27]:(5)$$L_{max}\leq \frac{C}{6f_{max}}$$
where C is the sound velocity in the medium, m/s; L is the acoustic grid length, m; f is the frequency of acoustic solution, Hz.

According to Equation (5), the higher the solving frequency, the denser the required acoustic grid. Therefore, a higher grid density is required to solve the acoustic field in the ultrasonic frequency band.

Based on the time-domain sampling theorem, the sampling interval should meet Equation (6) to recover the original signal according to each sampling value completely. In other words, the sampling frequency should be more than twice the signal’s highest frequency [24,25]. Simultaneously, the time-domain sampling frequency should meet the requirements of Equation (7). According to Equations (6) and (7), the time interval and computation duration of the transient solution of the time-domain flow field can be obtained by referring to the highest frequency and the frequency resolution of the sound field to be solved:(6)$$\Delta t\leq \frac{1}{2f_{max}}$$
(7)$$N=\frac{1}{\Delta f\Delta t}$$
where Δf is the frequency resolution, N is the time-domain sampling frequency, Δt is the time-domain sampling interval, and NΔt is the time-domain sampling period.

### 2.3. Computational Model

#### Flow Field Sensitivity Analysis

Based on the Fluent program, the flow fields under 15 groups of working conditions with different leakage hole sizes, different leakage hole shapes, and different annular pressures are computed, as shown in Table 2.

The computational domain involves the whole annulus region. Since fluid computation requires high grid accuracy and the size of the computational model is large, ICEMCFD (Integrated Computer Engineering and Manufacturing code for Computational Fluid Dynamics) is used to plot the structured grid. ICEMCFD (version of 2020 R1) is a professional CAE pre-processing software. In the radial direction, the grids of the boundary layers of the inner and outer sides of the annulus region are densified. As shown in Figure 2, the grids around the leakage holes are also densified. The turbulence model is based on the standard k-e model [28]. In Fluent, the medium is set as air; the upper and lower end faces of the annulus region are set as pressure inlet and pressure outlet, respectively, and the pressure value is set as annular pressure; the shape of the leakage hole is set as a square shape, the outlet of the leakage hole is set as pressure outlet, and the pressure value is set as 0.1 MPa.

Four sets of grids with different grid densities are computed respectively to verify the grid independence. The computation results mainly examine the outlet flow rate at the leakage outlet. As shown in Table 3, when the number of grids increases from 760,000 to 890,000, the flow rate at the leakage outlet only increases by 0.8 m/s (0.2%). Considering the computational efficiency, grids with 760,000 nodes were adopted in the following computation.

### 2.4. Sound Field Computation Based on the Equivalent Sound Source

The acoustic computation is carried out on a typical working condition with a square leakage hole of 0.4 mm in side length and an annular pressure of 10 MPa. The steps include flow field computation, equivalent sound source computation, structural mode computation, and acoustic vibration coupling computation. As shown in Figure 3, the flow field computation adopts a small-size model near the leakage hole, and the sound source and sound propagation zones are coupled through the sound pressure at their interface.

The dimensions of the sound source area (near the leakage hole): the left side of the sound source area is a hemisphere with a diameter of 50 mm, and the right side of the propagation area is 30 mm in diameter and 100 mm in length. the middle part is 5 mm long and 0.5 mm wide (Figure 3a). Sound propagation zone model (annulus region): 215 mm outer diameter, 109 mm inner diameter, 200 mm length, leakage position for the 50 mm diameter hemisphere (Figure 3b). Pipe model size: outer diameter 220, inner diameter 104 mm, thickness 5 mm (Figure 3c). Exterior zone model for sound propagation (oil casing pipe): diameter 99 mm, length 200 mm (Figure 3d).

#### 2.4.1. Flow Field Computation

A small-size model is adopted for flow field computation, and the boundary conditions and physical property parameters are set according to Section 2.3. Transient computation is carried out to obtain the flow field distribution in the time domain based on steady-state computation convergence, which is used as the source term of sound field computation. According to Equations (6) and (7), and considering the frequency requirement of 20–40 kHz and the frequency resolution of 1000 Hz, the time step of flow field transient computation is set as 1.25 × 10^−5^ s, and a total of 80 steps are computed. After computation, the flow field distribution at different time points is derived to obtain the near-field quadrupole sound source.

#### 2.4.2. Equivalent Sound Source Computation

The near-field acoustic grid size is 0.5 mm (Figure 4), which satisfies the requirements of Equation (5). The air density is set as 115 kg/m^3^. The outer surface of the computational domain (excluding the surface in contact with the pipe wall) is set as a non-reflective boundary. Field points are set at the interface between the sound source zone and the sound propagation inner domain (annulus region). The solving frequency is 20–40 kHz, and the frequency interval is 1000 Hz. The sound field distribution of the sound source zone can be computed. Lastly, the sound pressure distribution at the interface can be derived as the external source term for the acoustic vibration coupling computation.

#### 2.4.3. Structural Mode Computation

The 2D shell element with a thickness of 5 mm and a grid size of 2 mm is used for pipe modal analysis. The material is 304 stainless steel, of which the elastic modulus is 2 × 10^11^ N/m^2^, the Poisson’s ratio is 0.34, and the density is 7800 kg/m^3^. The inlet and outlet ends of the model are simply-supported. Thus, the modal frequencies and mode shapes of the oil pipe and casing pipe can be computed.

#### 2.4.4. Acoustic Vibration Coupling Computation Based on the Structural Mode

The medium in the outer domain of the acoustic propagation zone (oil pipe zone) is methane, and the medium in the inner domain of the acoustic propagation zone (annulus region) is pressurized air. The upper and lower end faces are set as non-reflective boundaries. The sound pressure distribution obtained from equivalent sound source computation is imported as the external source term. The structural modes are imported, the solving frequency is set within 20–40 kHz with a frequency interval of 1000 Hz, and five field points are set at the central axis of the oil pipe. Then, the acoustic vibration coupling computation based on structural modes can be performed in the LMS Virtual LAB program to obtain the sound field distribution. The Acoustic grids in the sound propagation area is shown in Figure 5.

## 3. Results and Discussion

### 3.1. Flow Field Sensitivity Analysis

The flow field distribution of a square leakage hole with a side length of 0.4 mm under different inner-outer pressure differences of 5, 10, 20, 30, and 45 MPa is shown in Figure 6. According to the flow field distribution, the flow field changes caused by leakage are mainly concentrated near the leakage hole, which has a minor influence on the pipeline interior.

Based on the different leakage hole shapes, sizes, and annular pressures provided in Table 2, the trend of maximum flow under each set of parameters is calculated. This is shown in Figure 7. According to the curve A, with an increase in annular pressure, the maximum flow rate of the leakage hole increases for the same leakage hole. However, this increasing trend gradually slows down. According to the curve A and B, the gas flow rate increases with the leakage hole size once it enters the hole. According to the curve A and C, the circular leakage hole has a higher flow rate than the square leakage hole for the leakage holes with the same cross-sectional area. The circular leakage hole has a larger wet cycle and a smaller hydraulic diameter, which leads to a larger resistance loss along the flowing path.

### 3.2. Sound Field Computation Based on the Near-Field Equivalent Sound Source

The acoustic source zone’s sound pressure level (SPL) cloud diagram is shown in Figure 8. It can be observed that the sound pressure at the interface between the sound source zone and the inner domain (annulus region) of the sound propagation zone is uniformly distributed. Therefore, it is reasonable to select the sound pressure of an arbitrary point at the interface as the external source for the computation of the sound propagation zone. The sound pressure distribution of the equivalent sound source is shown in Figure 9. When the frequency is 32,000 Hz, the highest sound pressure amplitude is 18 Pa. Figure 10 shows the structural modal vibration shapes of the oil and casing pipe. At the frequency of 20,000.6 Hz, the maximum displacements of the oil and casing pipe are 40 and 27.7 mm, respectively.

The sound pressure distribution in the annulus region and inside the oil pipe at different frequencies is shown in Figure 11. Since the sound pressure in the annulus region is much higher than that inside the oil pipe, the sound pressure distribution in the oil pipe shows no obvious change in the sound pressure diagram. Figure 12 shows the SPL distribution. It can be seen that the sound pressure at the position (inside the oil pipe) directly facing the leakage hole is higher than that at other positions. Consequently, the ultrasonic testing equipment can detect the leakage sound field signal more easily when placed in this position.

The spectrum curves of the sound pressure frequency of five field points (A, B, C, D, and E) on the axis line of the oil pipe are shown in Figure 13. The acoustic wave characteristics are consistent at different positions in the axial direction, and the sound pressure shows no obvious difference. There are several peak values on the frequency spectrum curve of each field point, most of which are concentrated in the range of 25,000–35,000 Hz. At 26,000 Hz, the peak value of the frequency spectrum takes the largest value of 0.309 Pa, i.e., 83.8 dB (the reference value is 2 × 10^−5^ Pa), which is significantly lower than that of the equivalent sound source.

## 4. Conclusions

In this paper, aiming at the flow field and sound field simulation of leakage detection of casing pipes, a sensitivity analysis on the leakage flow field was conducted, and a set of simulation models for the sound field in oil pipes for test computation was established. Conclusions can be drawn as follows:The equivalent sound source model can be effectively used for sound field calculation of casing leakage, and sonoacoustic-vibroacoustic coupling calculation was carried out in LMS Virtual LAB. Compared with full-scale sound source modeling, the equivalent sound source method can reduce the flow field calculation time and lower the flow field and sound field coupling calculation time, improving the analysis efficiency when used for larger-scale analysis and calculation.According to the numerical simulation of the micro-leakage flow field of the casing, the fluid flow rate at the leakage hole increases with internal and external pressure differences and the area of the leakage hole. The flow rate at the round leak hole is higher for the same leak hole area than at the square leak hole.According to the coupled analysis of acoustic vibration, when the frequency is 32,000 Hz, the amplitude of sound pressure is maximum. The sound pressure is greatly reduced once the sound wave caused is transmitted via casing leakage to the tubing through the tubing wall. Moreover, the sound pressure in the tubing near the sound source is relatively large. Hence, it is considered that the leakage point occurs at the sound field signal of the leakage detection, where the sound pressure is relatively high.

## Figures and Tables

**Figure 1 materials-16-00386-f001:**
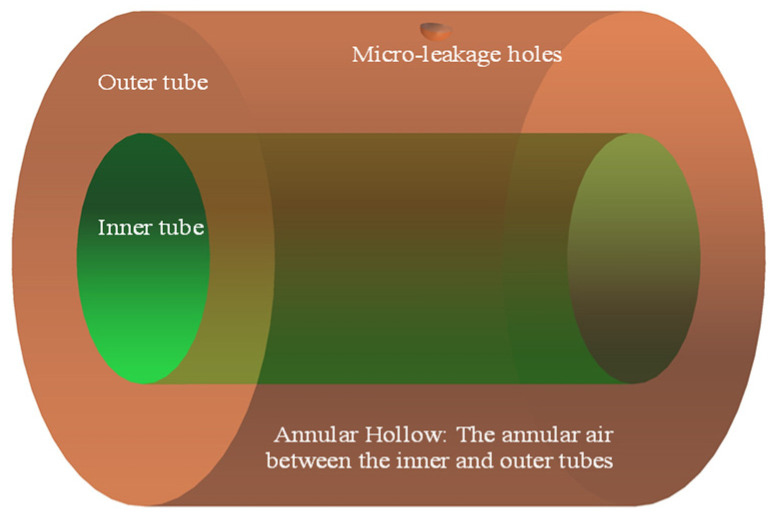
Schematic of micro-leakage of casing pipes.

**Figure 2 materials-16-00386-f002:**
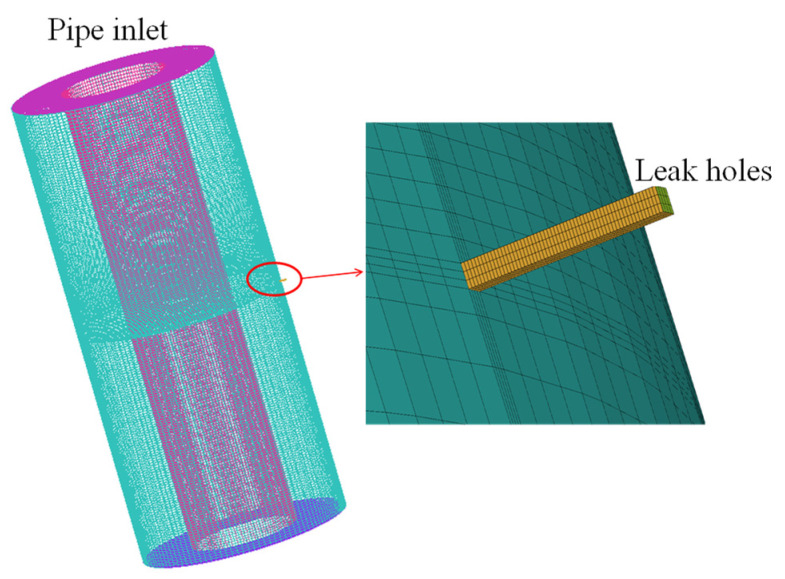
Grid partition overview.

**Figure 3 materials-16-00386-f003:**
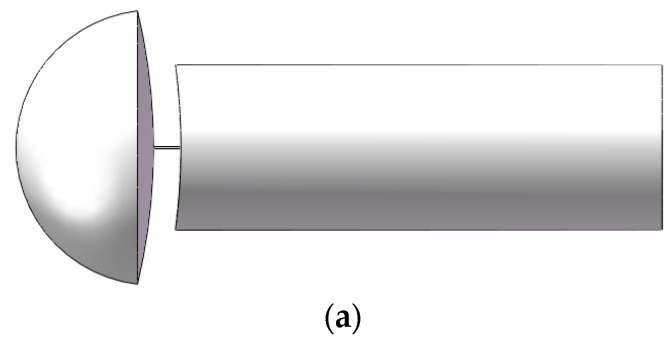
Schematic diagram of computational domains. (**a**) sound source zone model (near the leakage hole). (**b**) sound propagation zone model (annulus region). (**c**) pipe model. (**d**) exterior zone model for sound propagation (oil casing pipe).

**Figure 4 materials-16-00386-f004:**
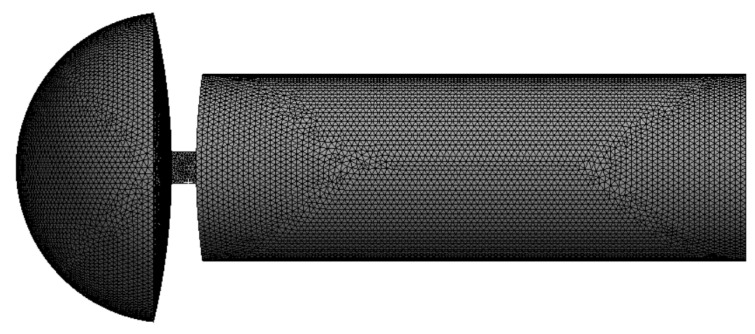
Acoustic grids in the sound source zone.

**Figure 5 materials-16-00386-f005:**
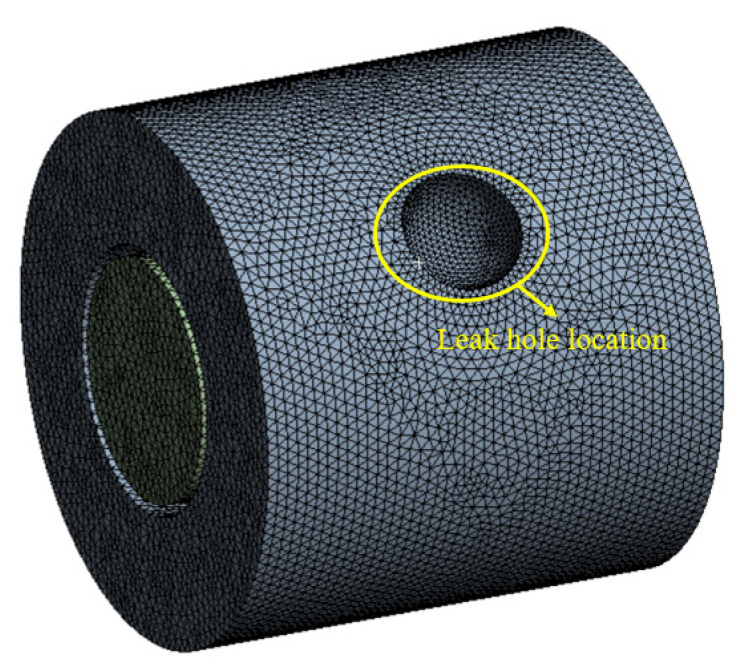
Acoustic grids in the sound propagation zone.

**Figure 6 materials-16-00386-f006:**
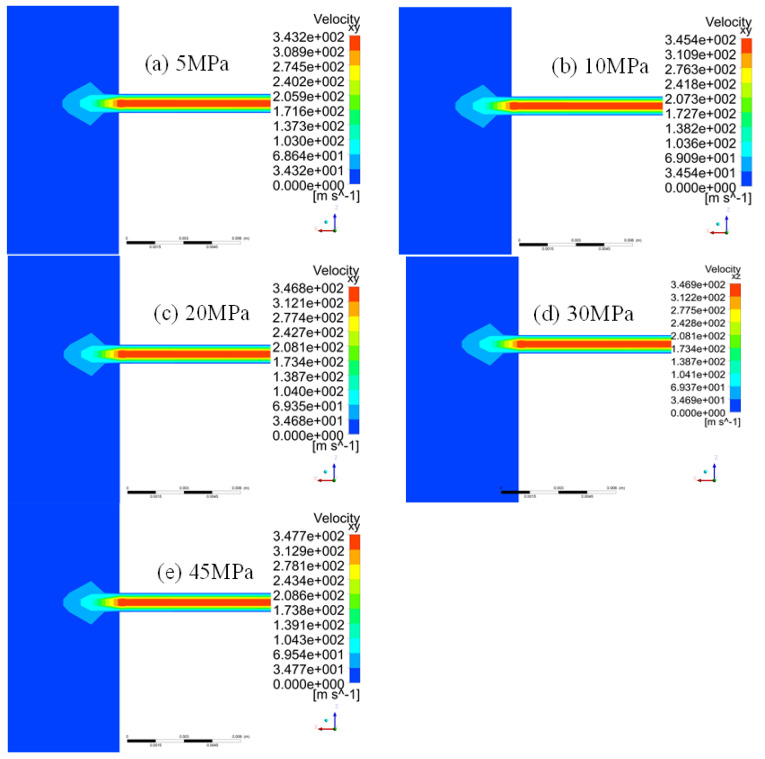
Flow field distribution of a square leakage hole with a side length of 0.4 mm under different inner-outer pressure differences.

**Figure 7 materials-16-00386-f007:**
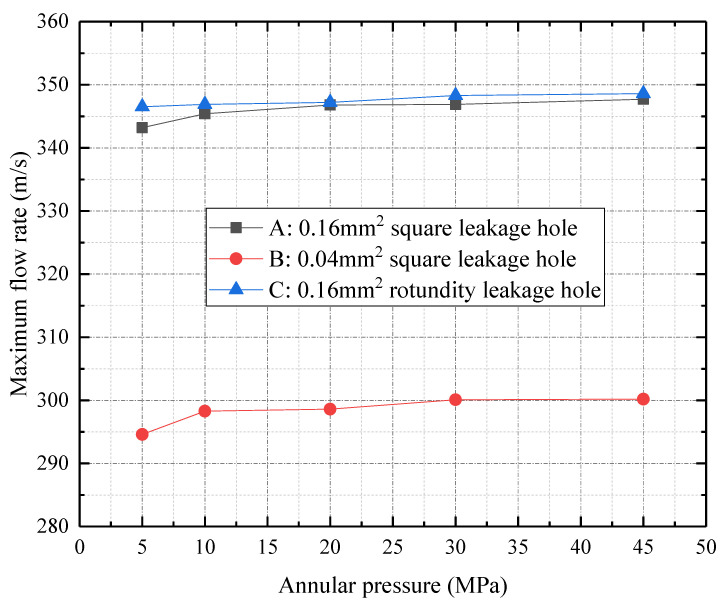
Maximum flow rate under different leakage hole shapes, sizes, and annular pressures.

**Figure 8 materials-16-00386-f008:**
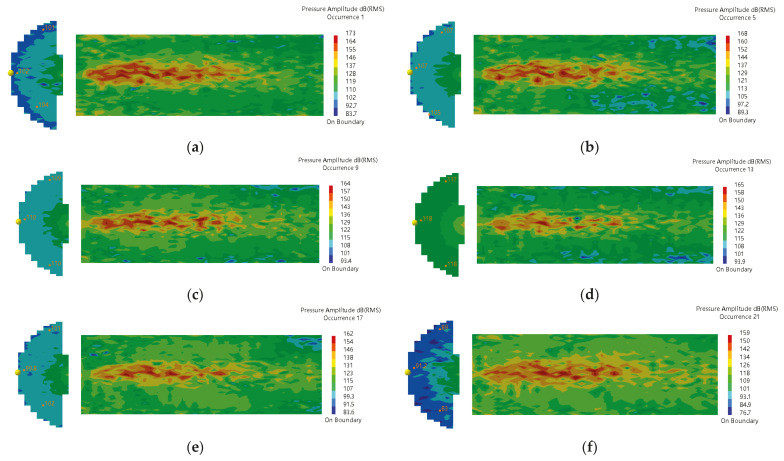
SPL distribution in the sound source zone. (**a**) 20,000 Hz (**b**) 24,000 Hz (**c**) 28,000 Hz (**d**) 32,000 Hz (**e**) 36,000 Hz (**f**) 40,000 Hz.

**Figure 9 materials-16-00386-f009:**
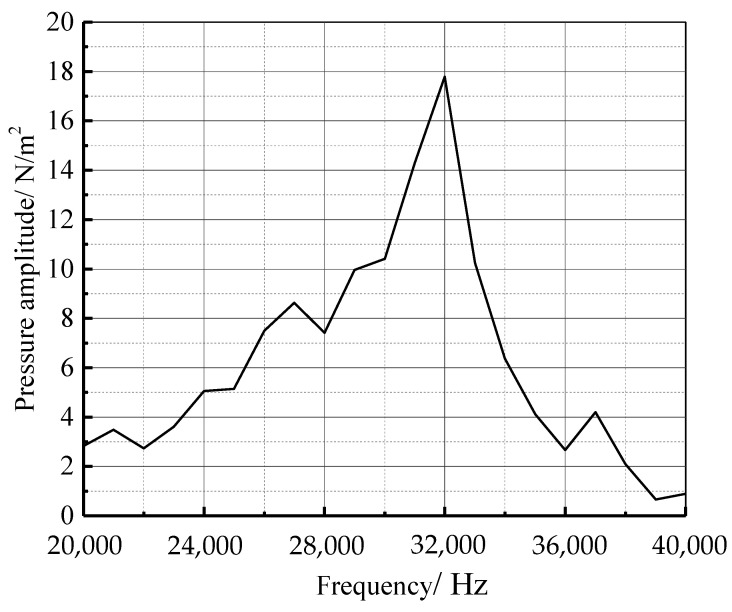
Sound pressure frequency spectrum curves of the equivalent sound source.

**Figure 10 materials-16-00386-f010:**
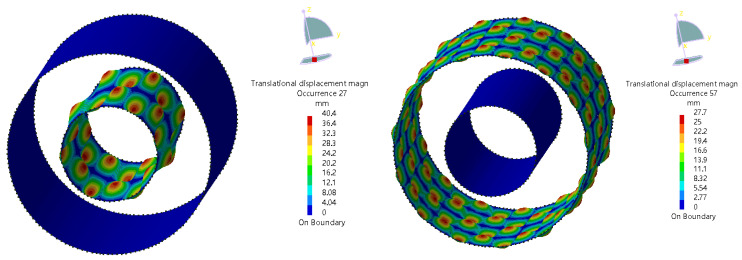
Structural modal vibration shapes of oil and casing pipe (20,000.6 Hz).

**Figure 11 materials-16-00386-f011:**
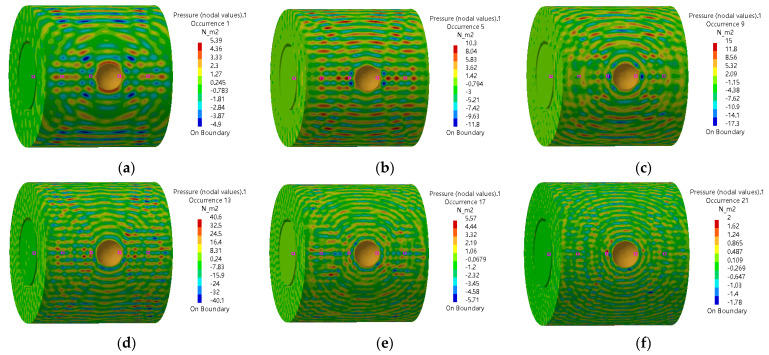
Sound pressure distribution in the annulus region and inside the oil pipe. (**a**) 20,000 Hz (**b**) 24,000 Hz (**c**) 28,000 Hz (**d**) 32,000 Hz (**e**) 36,000 Hz (**f**) 40,000 Hz.

**Figure 12 materials-16-00386-f012:**
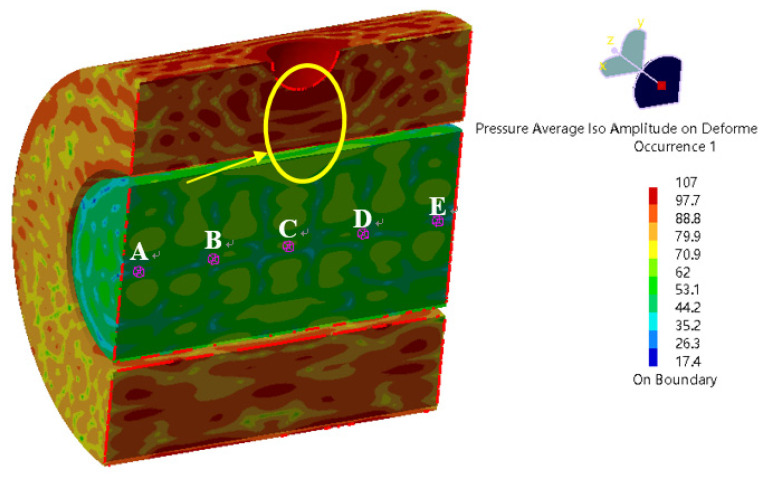
SPL distribution in the annulus region and inside the pipeline (20,000 Hz).

**Figure 13 materials-16-00386-f013:**
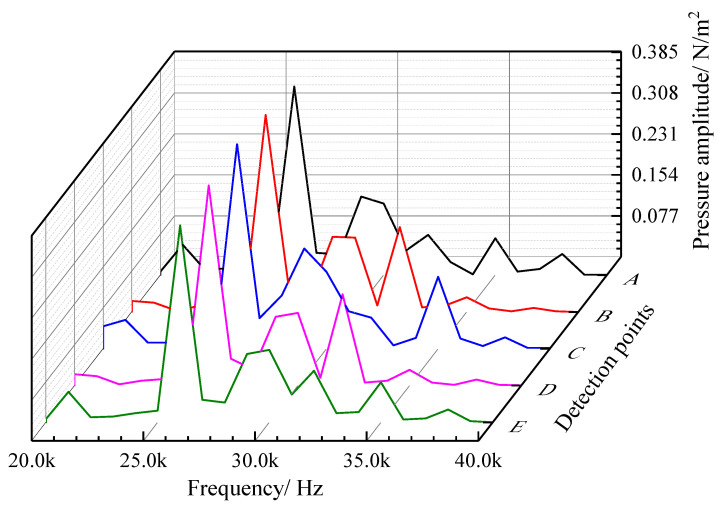
Sound pressure frequency spectrum curves of the five sampling points on the axis line of the oil pipe.

**Table 1 materials-16-00386-t001:** Parameter table of the physical model.

Parameter	Value
The inner diameter of oil pipes, mm	104
Wall thickness of oil pipes, mm	5
The outer diameter of casing pipes, mm	220
Axial length, mm	200
Medium inside oil pipes	Methane
Medium inside casing pipes (annular medium)	Air
Environment medium outside leakage holes	Air
Environment pressure outside leakage holes, MPa	0.1
Structural material for pipeline	304 stainless steel

**Table 2 materials-16-00386-t002:** Computation of working conditions for flow field sensitivity analysis.

Computation Example	Leakage Hole Shape	Leakage Hole Size /mm^2^	Annular Pressure /MPa
1	square	0.16	5
2	square	0.16	10
3	square	0.16	20
4	square	0.16	30
5	square	0.16	45
6	square	0.04	5
7	square	0.04	10
8	square	0.04	20
9	square	0.04	30
10	square	0.04	45
11	rotundity	0.16	5
12	rotundity	0.16	10
13	rotundity	0.16	20
14	rotundity	0.16	30
15	rotundity	0.16	45

**Table 3 materials-16-00386-t003:** Grid independence analysis.

Grid Number (Ten Thousand)	Flow Rate at the Leakage Outlet (m/s)
53	300.5
62	320.1
76	343.2
89	344.0

## Data Availability

Data sharing not applicable. No new data were created or analyzed in this study. Data sharing is not applicable to this article.

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
