# Peer review of "Numerical Study of Casing Microleakage Flow Field Sensitivity and Acoustic Field Characteristics"

_materials, 2022, doi:10.3390/ma16010386_

Round 1

Reviewer 1 Report

This paper performs numerical simulation on the leakage phenomenon of casing pipes in the field of oil transportation. Using a commercial software, the influencing factors such as the leakage hole sizes, shapes, and inner-outer pressure difference are investigated. Moreover, specifying the equivalent sound source, the authors analyzed the sound field in oil pipes. As a result, it is found that the circular leakage hole had a larger flow rate, and that the ultrasonic signal can be detected easily in the vicinity of the sound source.

Although the manuscript is readable, the novelty is not clear. The originality of this work should be emphasized more. For example, what is the difference from the reference [16]? I could not get the article, but, in the line 14 of page. 2, the following statement is found. “Zhang et al. [16] analyzed the influences of the shape of leakage holes, the size of leakage holes, the internal pressure of oil casing pipes, and the annulus medium on the leakage flow fields. According to their results, the circular leakage hole had the largest leakage flow rate.” The finding seems to be the same. In addition, the findings of the ease of the close sound source seems to be not surprising and trivial. It is expected that more significant findings can be obtained by further investigation based on the developed models. I think that the detectable hole size and the detectable distance from the sensor are interesting from the practical point of view.

The content of this research may be out of the scope of the section “porous materials”. Other articles such as “Acoustics” of MDPI and some journals of nondestructive inspection may be suitable.

General comments and questions are follows.

1.       The explanation of Eq.(1) is not sufficient. What is “f”? What is “c_0”? What is “p_0”?

2.       Where is the point of maximum flow in Figure 6?

3.       In Figure 8, why the peak is 32000 Hz? Does the peak frequency change for the different hole shape and size?

Minor comments are follows.

Eq. (7) is the same as Eq. (6). Correct it.

Table 3, Figure 4, and Figure 5 are not mentioned in the text.

Explain the word “ICEM” in the second sentence of the last paragraph in page 5.

In the second line of subsection 2.4.1, is “the subsection 3.1” correct?

In page 9, correct “Table 1” to “Table 4”.

Author Response

Dear Reviewer
We would like to thank you for giving us useful hints, criticism and suggestions that helped us to improve the manuscript. Here we submit the revised manuscript. We followed your suggestions and also performed a complete language revision by a native specialist. All the changes indicated (in red) in the revised manuscript.
Sincerely yours,
Jifang WAN,

Reviewer 2 Report

In this paper, the authors investigate the influencing factors of the leakage flow field by considering the results of numerical simulations. The micro-leakage flow field and sound field of casing pipes were analyzed by varying the working conditions of the leakage holes. In particular, it was shown that the pressure fluctuations inside and outside of the pipes strongly affect the flow rate. Moreover, the acoustic wave is greatly reduced after passing through oil pipes. The developed FEM model is of a particular interest for industrial applications because the leakage of pipes significantly affects the safety and rate of oil and gas transportation.

The manuscript is well written in terms of scientific content, however, a more comparative analysis and description regarding the obtained results is required. Some general comments are as follows:

1. References to equations (1-4) should be added to the text.

2. Please, optimize the number of columns in Table. 2. For clarity, it is also suggested to perform circular calculations for a 0.2 mm - diameter hole and give the data of the hole size in mm2. Currently, the influence of the hole shape on the flow rate seems to be incomplete.

3. What reason could be attributed to insignificant change of flow field velocity at the hole while the pressure varies from 5 up to 45 MPa? It is probably worth considering a much lower pressure level. Could the authors discuss about this peculiarity in the sense of practical effects?

4. There is a typo in the enumeration of the Table 4 (Line 266); however, it is worth noting that the data in Table 2 is duplicated in Table 4. It is suggested to delete Table 4, while the simulation results could be converted into a graph.

5. Line 211: What is the reason for such a wide frequency step of 1000Hz? The time of flow computation seems to be quite fast.

6. On what basis was the sound pressure amplitude in the Fig.8 calculated? It’s seen from Fig.7, that the maximum for both absolute and relative pressure (dB) corresponds to the frequency of 20 kHz.  Therefore, it is not clear why "the sound pressure takes the highest amplitude value at the 32 kHz". How complicated would be the signal in case of multiple holes?

7. Fig. 9: How exactly are these modal shapes evaluated? What was the boundary conditions? Why did you choose a frequency of 20000.6 Hz for the calculation? It has been previously shown that the maximum sound pressure was measured at 32 kHz.

8. The author do not describe the future research plan in the conclusion part. The conclusion is just a summary of the article without any information about research prospects.  

Author Response

(The authors gave the same response as above.)

Reviewer 3 Report

The following comments have been made to improve the quality of the manuscript.

1)    The title needs revision. It should be in such a way that it should depict the exact work. A good title contains the fewest possible words that adequately describe the contents of your research work.

2)    Lines 12 and 14 should be revised (On the other hand).

3)    While reading the abstract, I didn’t see the exact objective of the work. Please be concise with the objective.

4)    Further, a sufficient description of the simulation methodology and outcome should be in the abstract.

5)    Please mention the specific outcome  (such as in percentage, number, temperature, etc) at the end of the abstract.

6)    Line 41 to 48 is not phrased well.

7)    The literature review is written well but the research gap is not mentioned clearly.

8)    The last passage of the introduction needs rephrasing. Please check lines 92-93.

9)    Where are CFD governing equations and boundary conditions etc?

10)        Where are geometric details/dimensions in the figure?

11)        What type of mathematical model was used?

12)        How the choice of turbulence model was made? You can get some benefits from this work: https://doi.org/10.2355/isijinternational.54.2578

13)        Did you study the grid independence test?

14)        Did you validate the numerical model?

15)        Please check the error in line 261.

16)        What methodology was utilized in inclusion entrapment at the surface level?

17)        Figure 6 is not clear.

18)        Did you use user-defined functions or any kind of mathematical code to configure flow and inclusion motions?

19)        The conclusion is insufficient. You should highlight a few particular results of this work.

20)        There are many grammatical mistakes in this work.

Author Response

(The authors gave the same response as above.)

Round 2

Reviewer 1 Report

The manuscript was improved. Thank you for the revision.

Reviewer 2 Report

My comments have been addressed. The current manuscript is suggested for publication.

Reviewer 3 Report

Thanks